# Zebrafish as a Human Muscle Model for Studying Age-Dependent Sarcopenia and Frailty

**DOI:** 10.3390/ijms25116166

**Published:** 2024-06-03

**Authors:** Paula Aranda-Martínez, Ramy K. A. Sayed, José Fernández-Martínez, Yolanda Ramírez-Casas, Yang Yang, Germaine Escames, Darío Acuña-Castroviejo

**Affiliations:** 1Centro de Investigación Biomédica, Facultad de Medicina, Departamento de Fisiología, Universidad de Granada, 18016 Granada, Spain; ampaula@correo.ugr.es (P.A.-M.); josefermar@ugr.es (J.F.-M.); yolandaramirez@ugr.es (Y.R.-C.); gescames@ugr.es (G.E.); 2Instituto de Biotecnología, Parque Tecnológico de Ciencias de la Salud, Universidad de Granada, 18016 Granada, Spain; 3Instituto de Investigación Biosanitaria (Ibs. Granada), Hospital Universitario San Cecilio, 18016 Granada, Spain; 4Department of Anatomy and Embryology, Faculty of Veterinary Medicine, Sohag University, Sohag 82524, Egypt; ramy.kamal@vet.sohag.edu.eg; 5Key Laboratory of Resource Biology and Biotechnology in Western China, Ministry of Education, Faculty of Life Sciences and Medicine, Northwest University, Xi’an 710069, China; yang200214yy@163.com; 6Centro de Investigación Biomédica en Red de Fragilidad y Envejecimiento Saludable (CIBERFES), Instituto de Salud Carlos III (ISCIII), 28029 Madrid, Spain; 7UGC de Laboratorios Clínicos, Hospital Universitario San Cecilio, 18016 Granada, Spain

**Keywords:** zebrafish, aging, skeletal muscle, mitochondria, sarcopenia

## Abstract

Currently, there is an increase in the aging of the population, which represents a risk factor for many diseases, including sarcopenia. Sarcopenia involves progressive loss of mass, strength, and function of the skeletal muscle. Some mechanisms include alterations in muscle structure, reduced regenerative capacity, oxidative stress, mitochondrial dysfunction, and inflammation. The zebrafish has emerged as a new model for studying skeletal muscle aging because of its numerous advantages, including histological and molecular similarity to human skeletal muscle. In this study, we used fish of 2, 10, 30, and 60 months of age. The older fish showed a higher frailty index with a value of 0.250 ± 0.000 because of reduced locomotor activity and alterations in biometric measurements. We observed changes in muscle structure with a decreased number of myocytes (0.031 myocytes/μm^2^ ± 0.004 at 60 months) and an increase in collagen with aging up to 15% ± 1.639 in the 60-month group, corresponding to alterations in the synthesis, degradation, and differentiation pathways. These changes were accompanied by mitochondrial alterations, such as a nearly 50% reduction in the number of intermyofibrillar mitochondria, 100% mitochondrial damage, and reduced mitochondrial dynamics. Overall, we demonstrated a similarity in the aging processes of muscle aging between zebrafish and mammals.

## 1. Introduction

The life expectancy of the population is significantly increasing, and by 2030, one out of six people worldwide will be 60 years old or older, whereas the population aged 60 years or over will double by 2050, and the population aged 80 years or over will triple. Aging is a major risk factor for numerous diseases, including sarcopenia, which is characterized by the progressive loss of skeletal muscle mass, strength, and function, leading to functional impairment, frailty, and increased mortality [1]. Thus, it is reasonably expected that there will be an increase in sarcopenia and sarcopenia-related deaths in the next years. Although the exact mechanisms of sarcopenia are not fully understood, it is known to involve alterations in muscle structure and function, reduced regenerative capacity, oxidative stress and inflammation, and mitochondrial alterations [2].

Although we are not aware of any studies that provide an established treatment or definitive diagnosis for this condition, preventive approaches, including exercise and increased protein intake, have been recommended [3]. Therefore, the use of improved and diverse animal models is imperative for gaining a comprehensive understanding of its pathogenesis. While rodents are the most used models for studying sarcopenia, alternative models such as *Drosophila* and *C. elegans* have been proposed because of their cost-effectiveness and shorter lifespans. However, it should be noted that these alternative models differ more from human skeletal muscle. Here, the zebrafish emerges as a novel and promising model to consider [4].

We have previously demonstrated in an aged mouse model that there is a reduction in type II muscle fibers, muscle fiber hypertrophy, and collagen infiltrations, which contribute to the loss of muscle mass and function [5]. Additionally, we observed a decreased mitochondrial number, in addition to mitochondrial damage such as disrupted cristae and swelling. There was also a reduction in lactate content and an increase in apoptotic nuclei [6]. Subsequently, we identified the pivotal role of the nucleotide-binding domain, the leucine-rich-containing family, and the pyrin domain–containing-3 inflammasome (NLRP3) in the development of sarcopenia [7]. More recently, we investigated the connection between sarcopenia and chronodisruption, specifically focusing on the loss of aryl hydrocarbon receptor nuclear translocator-like protein 1 (Bmal1) [8], as it plays a crucial role in the maintenance and repair of skeletal muscle [9,10]

Zebrafish have emerged as a promising new model for studying human diseases because of their numerous advantages, including their small size, rapid and extrauterine embryonic development, embryo transparency, and ease of genetic manipulation, resulting in a more economical model than mammalian models. Additionally, it shares a great resemblance to the human genome (71% of genes) and significant similarity in cellular processes of development, aging, and death [11], making it also a good model for studying aging. The zebrafish has a maximum lifespan of 5 years under laboratory conditions [12] and is considered an adult at 3 months of age when it reaches sexual maturity [13]. Like humans, zebrafish experience gradual senescence [14,15], increased oxidative stress [16], mitochondrial damage [17], skeletal muscle degeneration [18], and decreased physical capacity during aging [19]. Notably, zebrafish skeletal muscle exhibits striking similarities to human muscle in both histological and molecular aspects [20], and additionally, exercise can reduce age-related sarcopenia [17,21]. Consequently, we advocate the use of zebrafish as a novel model to investigate the physiology of aging, particularly in relation to skeletal muscle, with the aim of establishing an effective diagnostic protocol and treatment for sarcopenia.

## 2. Results

### 2.1. Morphometric and Motor Activity Analysis Revealed a Decline in Physical Condition and a Concurrent Increase in Frailty with Aging

As mentioned before, sarcopenia is characterized by the loss of muscle mass, strength, and function, leading to reduced physical capacity and frailty [1]. Therefore, we decided to analyze morphometric aspects and locomotor activity in the zebrafish to evaluate the degree of frailty. As expected, there was a progressive increase in BMI with age, which can be attributed to the increase in weight and length (Figure 1). Interestingly, the groups aged 2 and 10 months did not display significant differences in BMI (0.019 g/cm^2^ ± 0.004 and 0.025 g/cm^2^ ± 0.003, respectively), despite the 10-month-old group having greater weight and length, although an increasing trend was observed in the latter (Figure 1). Notably, the BMI was significantly higher in the 30-month-old fish than in the 2-month-old fish with a value of 0.029 g/cm^2^ ± 0.006, although it did not differ significantly from the 10-month-old group (Figure 1C). However, the total weight and total length of the 30-month-old group did present differences among both groups (Figure 1A,B). Finally, the 60-month group showed a significant increase in BMI, being 0.040 g/cm^2^ ± 0.016, which likewise corresponded to a significant increase in weight and length (Figure 1).

The locomotor activity recorded by the video-tracking open-field test did not reveal changes in the 2- and 10-month groups in terms of total distance (4667.738 cm ± 1386.530 and 5143.711 cm ± 1186.039, respectively) (Figure 1D), maximum speed (35.302 cm/s ± 11.625 and 31.961 cm/s ± 6.469, respectively) (Figure 1E), or mean speed (4.182 cm/s ± 1.017 and 4.607 cm/s ± 0.553, respectively) (Figure 1F). The total distance was significantly reduced in the 30- and 60-month groups, with values of 3164.341 cm ± 1420.143 and 2958.518 cm ± 723.492 for each one, compared with the 10-month group (Figure 1D). Furthermore, significant changes were observed in the maximum speed of the 60-month group compared with the 2- and 10-month groups which decreased to 18.552 cm/s ± 7.651 (Figure 1E). Lastly, age also influenced mean speed, which exhibited a significant reduction in the 30- and 60-month groups to 2.635 cm/s ± 1.183 and 2.464 cm/s ± 0.603 accordingly, contrasting with the 2- and 10-month groups (Figure 1F).

The frailty index is commonly employed in the clinic to assess the level of vulnerability of patients. Recently, this measure has been translated to primates and mice [6,22], and we have now adapted it to zebrafish. The aforementioned morphometric and locomotor measurements were used to calculate the frailty index (FI). Age exhibited an impact on the frailty index, with an increase of 0.125 ± 0.144 observed at 30 months of age compared with the 2- and 10-month groups, which had a value of 0, and this increase was statistically significant at 60 months with a level of frailty of 0.250 ± 0.000 (Figure 1G). You can find all the data in Appendix A.

### 2.2. The Structure and Organization of the Zebrafish Skeletal Muscle Are Significantly Compromised with Age

Next, we examined the structure of the skeletal muscle, which, as is known, undergoes degeneration during aging in humans, mice, and zebrafish [2,5,18]. Light microscopy images of longitudinal sections stained with hematoxylin and eosin (H&E) stain showed a normal structure and organization of zebrafish skeletal muscle fibers in the 2- and 10-month-old groups (Figure 2A), with no differences in the number of myocytes (0.086 myocytes/μm^2^ ± 0.016 and 0.072 myocytes/μm^2^ ± 0.0.009, respectively) (Figure 2B), but with an increase in myocyte width at 10 months (2 mo = 15.23 μm ± 2.055 and 10 mo = 21.65 μm ± 2.455) (Figure 2C). At 30 months, while there was still good organization of the fibers (Figure 2A), hypertrophy of myocytes was observed, which was related to a reduction in the number of myocytes (0.056 myocytes/μm^2^ ± 0.004) (Figure 2B) and an increase in the myocyte width (27.95 μm ± 3.209) (Figure 2C). In contrast, older fish exhibited disorganization of the skeletal muscle and an increase in connective tissue (Figure 2A), resulting in a value of 0.031 myocytes/μm^2^ ± 0.004, which was significantly decreased compared with the other groups (Figure 2B), and a significant narrower width (17.3 μm ± 2.026) compared with the 10- and 30-month groups (Figure 2C).

On the other hand, VG staining revealed low collagen content at 2 months (6.023% ± 0.713) and at 10 months (7.944% ± 1.308) (Figure 2D,E). In contrast, an age-related increase in collagen content was detected, with a slight significant increase at 30 months ata 9.272% ± 0.788 and a more substantial increase, i.e., 15.01% ± 1.639 of collagen content at 60 months (Figure 2D,E).

### 2.3. Pathways of Muscle Growth and Differentiation undergo Alterations with Aging in Zebrafish

Muscle protein synthesis is modulated by the Akt/mTOR/p70s6k pathway, which is affected by age in mammals [23,24]. We analyzed the gene expression of Serine/Threonine Kinase 1 (akt) and p70 ribosomal S6 kinase (p70s6k) to study how aging impacted this pathway in zebrafish skeletal muscle. Concerning akt, we observed an age-dependent reduction in its expression, which was significant in the 30- and 60-month groups, which had a value of around 0.4 ± 0.01 compared with the younger group (Figure 3A). Because p70s6k acts downstream of akt and is one of the final steps of the pathway [25], it exhibited a marked decrease of about 14 times lower in the 10- (0.068 ± 0.024) and 30- (0.074 ± 0.066) month groups, and 50 times less in the 60-month group (0.020 ± 0.016) than in the 2-month group (Figure 3B). 

In addition, we investigated whether aging had an impact on muscle growth and differentiation pathways in zebrafish. Myogenin, which plays a key role in muscle myogenesis [26], exhibited a progressive, age-dependent, and significant decrease by half in the 60-month group (0.429 ± 0.151) compared with the 2- and 10-month groups (Figure 3C). On the other hand, Beta-Interferon Gene Positive Regulatory Domain I-Binding (prdm1a) is a repressor of fast muscle fiber differentiation and a promoter of slow fiber differentiation [27,28]. We observed a tendency to increase its expression in the two oldest groups, although the difference was not significant (Figure 3D).

### 2.4. Zebrafish Aging Revealed Alterations in the Ultrastructure of Muscle and Mitochondria

In addition to skeletal muscle degeneration, mitochondrial dysfunction in sarcopenia has been described in the literature for both humans and zebrafish [17,29]. Electron microscopy examination of 2-month-old fish revealed the normal ultrastructure of the skeletal muscle (Figure 4A), which comprises longitudinally arranged myofibrils (Mf). However, some myofibrils were shown to be ill-developed, with disorganized my filaments and intermyofibrillar components (black asterisk). The intermyofibrillar spaces displayed a sarcoplasmic reticulum (SR), transverse tubules (T) forming a terminal cisterna (C), and intermyofibrillar mitochondria (IFM). Both IFM and subsarcolemmal mitochondria (SSM) were compacted and showed well-organized cristae. The nucleus (N) was observed in its usual peripheral position beneath the sarcolemma (black arrow). 

At the age of 10 months, the muscle was fully developed and characterized by well-aligned myofibrils and sarcomeres (Figure 4A). Transverse tubules were normally formed, the SR network was well organized, and the nucleus was positioned normally. The IFMs were compacted and exhibited organized cristae. However, individual cases revealed vacuolation (white arrow). Numerous small-sized membranous vacuoles (v) of possibly autophagic nature were observed among the SSM, which remained entirely normal and compacted. Less interstitial connective tissue (In) and glycogen droplets (g) are also depicted. 

The skeletal muscles of 30-month-old fish demonstrated organized myofibril and sarcomere alignment (Figure 4A). Hypertrophy was observed in both the IFM and SSM, with normally arranged cristae. However, few exhibited indistinct cristae (white asterisk). Numerous vacuoles and swelling of the SR and triads were evident. Moreover, the shrinkage of the peripherally positioned nucleus was observed.

At the age of 60 months, myofibril disorganization and adjacent sarcomere misalignment were demonstrated, with the presence of wide intermyofibrillar spaces and swelling of SR. Both IFM and SSM displayed abnormal structures and were entirely vacuolated with damaged and/or disorganized cristae (Figure 4A).

Morphometric analysis revealed a significant reduction in the length of sarcomeres with aging (10 mo = 1.888 μm ±0.098, 30 mo= 1.837 μm ± 0.069, 60 mo = 1.744 μm ± 0.098) compared with the value of 2.342 μm ± 0.3794 of 2-month-old fish (Figure 4B). The number of IFMs was 5.000 ± 0.817, 5.500 ± 0.577 and 4.250 ± 1.258 for the 2-, 10- and 30-month groups, and it was significative lower, 3.286 ± 1.380, in the 60-month group (Figure 4C). The cross-sectional area (CSA) of IFM decreased by around 50% in the oldest group (0.4139 μm^2^ ± 0.120) regarding the other groups (Figure 4D). However, the number of SSMs was significantly increased up to 8.286 ± 2.059 at the age of 30 months, whereas CSA showed non-significant differences (Figure 4E,F). Bands A increased significantly in 60-month-old fish (2 mo = 1.488 μm ± 0.044, 10 mo = 1.309 μm ± 0.122, 30 mo = 1.451 μm ± 0.053, 60 mo = 1.666 μm ± 0.255) (Figure 4G), similarly to the H band (2 mo = 0.105 μm ± 0.014, 10 mo = 0.099 μm ± 0.010, 30 mo = 0.113 μm ± 0.021, 60 mo = 0.146 μm ± 0.012) (Figure 4H). However, band I decreased significantly in the oldest group (2 mo = 0.488μm ± 0.219, 10 mo = 0.364 μm ± 0.069, 30 mo = 0.3729 μm ± 0.052, 60 mo = 0.300 μm ± 0.066) (Figure 4I). Finally, a pronounced increase in the percentage of damaged mitochondria in aged muscle was observed, reaching 100% ± 0.000 at 60 months (Figure 4J).

Through TEM analysis, we observed a decline in the number of mitochondria in the muscle with aging. Our investigation also delved into whether crucial processes for preserving mitochondrial homeostasis, such as mitochondrial dynamics, were affected. Mitochondrial dynamics include events of mitochondrial fission and fusion to determine the distribution of mitochondria within cells. Mitochondrial fusion requires the fusion of the outer mitochondrial membrane mediated by mitofusins (MFNs) and the fusion of the inner mitochondrial membrane carried out by Optic Atrophy 1 (OPA1). On the other hand, mitochondrial fission involves a constriction of the mitochondrial membrane by Dynamin-related protein 1 (DRP1) and its subsequent scission mediated by Dynamin 2 (DYN2). Fusion favors the exchange of contents between mitochondria, while fission contributes to the elimination of damaged mitochondria [30]. Regarding mitochondrial fusion, we observed a trend of decreasing expression of mfn1 with age, which was significantly 5 times lower at 60 months (0.179 ± 0.077) compared to the 2-month group (Figure 5A), just like opa1 (0.233 ± 0.104) (Figure 5B). Conversely, when examining mitochondrial fission, we found a slight increase in the expression of dyn2 (Figure 5C) and drp1 (Figure 5D) at 10 and 30 months, respectively. The dyn2 expression in the 60-month group (0.427 ± 0.160) was 2.5 times less than in the 2-month group, with the expression of drp1 (0.449 ± 0.174) mirroring the pattern observed with mitochondrial fusion.

### 2.5. Confocal Analysis of Skeletal Muscle in Mito-GFP Zebrafish Shows Consistent Loss of Mitochondria with Age

To confirm the mitochondrial damage, we performed confocal microscopy of the skeletal muscle of Mito-GFP zebrafish. Confocal analysis confirmed the TEM data, revealing the mitochondrial volume in the skeletal muscle of Mito-GFP zebrafish (Figure 6A). In the younger groups (2 and 10 months), we observed a high GFP intensity in the muscle, a 1.002 mean intensity/number of DAPI ± 0.325, and a 1.195 mean intensity/number of DAPI ± 0.32, respectively, indicating a higher number of mitochondria (Figure 6B). By the time the fish reached 30 months of age, there was a significant reduction in the number of mitochondria within the muscle (0.537 mean intensity/number of DAPI ± 0.245), and at 60 months, we observed a drastic reduction in the GFP signal (0.349 mean intensity/number of DAPI ± 0.183) (Figure 6B).

## 3. Discussion

With this study, we have demonstrated for the first time that the zebrafish is a good model for studying the deterioration of skeletal muscle with age. The weakening of the skeletal muscle reached its maximum at the age of 60 months, which represents the highest lifespan of zebrafish under laboratory conditions [12,31]. During the aging of zebrafish, we observed a progressive increase in BMI due to greater weight and length, which is consistent with the observation in humans that an increase in BMI correlates with a higher risk of sarcopenia [32]. Additionally, we also demonstrated an age-associated loss of motor function similar to that in humans, since sarcopenia involves the loss of muscle mass and function [33]. Using these morphometric and motor features, we proposed adapting for the first time the frailty index, which is commonly used to measure the degree of vulnerability in patients, primates, and mice [6,23]. The results obtained from the morphometric measurements and motor activity led to an increase in the frailty index with age. Hence, this index represents an initial, non-invasive step in the diagnosis of sarcopenia, which would subsequently require more specific and in-depth analyses. Our data support those of Rutkove et al., who used electrical impedance testing to detect muscle atrophy [18].

Concerning the skeletal muscle structure, previous studies reported a decrease in the cross-sectional area of muscle fibers and an increase in fibrosis in 21-month-old fish [22], whereas others detected D-galactose-induced sarcopenia in fish [21]. Here, we also demonstrated age-dependent skeletal muscle atrophy. The number of myocytes and sarcomere length decreased with age, muscle ultrastructure was affected, and we observed muscle hypertrophy at 30 months, which preceded the loss of mass, along with a significant increase in collagen content at 60 months of age.

Muscle atrophy involves an imbalance in the protein degradation–synthesis process, with an excess of degradation [34]. In a study with 21-month-old fish, the authors observed a decrease in the IGF1/PI3K/Akt/mTOR protein synthesis signaling pathway and, conversely, an increase in the expression of genes and proteins associated with protein degradation, such as Murf, Fbxo32, FoxO, and the TRIM family, among others [22]. We then studied one of the most important and initial genes in this pathway, Akt, and by contrast, p70S6, which acts downstream, directly influencing protein synthesis [35]. We found a significant reduction in the expression of both genes with age, which correlates with the muscle atrophy found in histological analysis. This finding suggests that there is a decrease in the signaling of this pathway, including the reduction in the expression of genes such as mTOR, 4EBP1, etc., and an increase in degradation genes, generating an imbalance between protein synthesis and degradation. Furthermore, the TNF-α factor and activation of the NFκB pathway during aging also activate protein degradation. Furthermore, the TNF-α factor and activation of the NFκB pathway also activate protein degradation [36], revealing the role of inflammation in muscle atrophy, as demonstrated elsewhere [37,38,39]. 

Muscle atrophy was caused not only by the disruption of the IGF1/PI3K/Akt/mTOR pathway signaling but also by the observed decrease in myogenin expression in aged fish, which also contributes to muscle degeneration. Myogenin belongs to the family of myogenic regulatory factors (MRFs), which are essential for muscle proliferation and differentiation. Specifically, myogenin is responsible for the fusion of myoblasts and their differentiation in both embryonic muscle development and adult muscle regeneration [40]. Recently, it has been shown in zebrafish that myogenin controls the number of resident muscle stem cells (MuSCs), their location in the muscle, the rate of active and inactive MuSCs, and the proper size of myofibers to maintain adult skeletal muscle homeostasis [27]. On the contrary, myostatin limits the development and growth of skeletal muscle, so its expression should be increased in our aged zebrafish model [41]. 

Parallel to the degeneration and loss of muscle mass during sarcopenia, there is a loss of function due to the replacement of fast muscle fibers responsible for muscle strength, with slow fibers involved in endurance [42]. Prdm1a is a transcription factor that represses sox6, promoting the differentiation of slow muscle fibers, and it also directly represses the transcription of specific fast fiber genes [43]. We have observed an increase in the expression of prdm1a in 30- and 60-month-old fish, probably indicating a higher repression of sox6. As already observed, the absence of sox6 in zebrafish promotes the differentiation of slow fibers [29].

The bioenergetics of skeletal muscle depend on mitochondria, and mitochondrial dysfunction occurs during aging, which affects the oxidative capacity of the muscle [44]. In a model of D-galactose-induced sarcopenia in zebrafish, as well as in fish aged up to 21 months, they observed an increase in ROS in the muscle, as well as a decrease in the expression of antioxidant factors such as SOD, GPx, and Nrf2 [21,22]. Since the mitochondria are the main source of ROS [45], this seemed to indicate mitochondrial dysfunction, which is indeed what they found: vacuolization, swelling, loss of mitochondrial cristae, rupture, a decrease in the number of mitochondria, and reduced mitochondrial respiration [21,22]. Our 60-month-old fish also showed a reduction in the number of IFM and their CSA, loss of cristae, vacuolization, and a 100% increase in mitochondrial damage. Furthermore, mitochondrial dynamics and biogenesis are essential for maintaining mitochondrial homeostasis [44]. In the two zebrafish sarcopenia models mentioned earlier, there was an alteration in mitochondrial fusion and fission events and a reduction in the AMPK/SIRT1/PGC-1α biogenesis pathway [21,22]. We decided to analyze mitochondrial dynamics since this process controls the distribution of mitochondria within the fibers and is so important because IFM mitochondria are organized in a reticulum, thanks to fusion and fission events, to facilitate energy distribution [44]. The expressions of mitochondrial fusion and fission genes in older fish were significative lower. Surely mitochondrial biogenesis is impaired in our model, just as the other authors found.

Finally, mitochondria work together with SR to maintain Ca^2+^ homeostasis in skeletal muscle, which is crucial for both function and intracellular signaling. Moreover, an increase in Ca^2+^ within mitochondria leads to an increase in ROS [44]. Our results revealed mitochondrial damage and SR swelling with aging, which likely indicates an alteration in muscle Ca^2+^ metabolism.

Therefore, we demonstrated for the first time the physiopathology of sarcopenia in heavily aged fish, which resembles the characteristics of the disease seen in mammals. However, studies on sarcopenia in zebrafish are scarce. The primary limitation of our model and other models is the extended duration of the study. Nonetheless, we propose that zebrafish offer a valuable model for studying the mechanisms of skeletal muscle aging because of their significant advantages. They can absorb substances from water through their skin and gills, thus facilitating the use of induced sarcopenia models such as the dexamethasone model and the chronic alcohol model [46]. This also enables the testing of numerous drugs for sarcopenia treatment, which reflect an environmental model of disease. Nevertheless, the primary advantage that makes zebrafish highly useful for research is their ease of mutagenesis. Thanks to the CRISPR/Cas9 system, multiple genetic models of sarcopenia could be generated, allowing the study of several genes as potential therapeutic targets [46].

## 4. Materials and Methods

### 4.1. Fish Maintenance and Experimental Groups

Mito-GFP adult zebrafish (*Danio rerio*) were provided by Kim et al. [47], and AB-strain adult zebrafish were provided by ZFBiolabs S.L (Madrid, Spain). The fish were maintained at the University of Granada’s facility at a constant water temperature of 28.5 ± 1 °C, under a photoperiod of 14 h of light and 10 h of darkness (with lights turning on at 08:00 a.m.), using a recirculation aquaculture system provided by Aquaneering Incorporated (Barcelona, Spain). Fish feeding, breeding, maintenance, and anesthesia procedures adhered to established published protocols [48]. For age-dependent studies, we used zebrafish of 2, 10, 30, and 60 months of age. AB-strain zebrafish were used for the analysis of motor activity, frailty index, transmission electron microscopy, and optical microscopy. In addition, Mito-GFP zebrafish, characterized by EGFP-labeled cytochrome c oxidase (COXVIII), were used for confocal microscopy analysis.

All experiments were performed according to the National Institutes of Health Guide for the Care and Use of Laboratory Animals, the European Convention for the Protection of Vertebrate Animals used for Experimental and Other Scientific Purposes (CETS # 123), and the Spanish law for animal experimentation (R.D. 53/2013). The protocol was authorized by the Andalusian Ethical Committee (#29/05/2020/068).

### 4.2. Assessment of Locomotor Activity

The zebrafish were individually placed in a tank with aquarium system water and acclimated for four consecutive days before the experiment was conducted on the fifth day. Locomotor activity was recorded over a 20 min period, with a preceding 3 min adaptation stage, using a digital video tracking system consisting of a CCD camera connected to a computer. The acquired images were processed using SMART 3.0 software (Panlab Harvard Apparatus, Barcelona, Spain). The distance traveled (cm), mean speed (cm/s), and maximum speed (cm/s) of each fish were quantified. 

### 4.3. Frailty Index Calculation

A clinical frailty index (FI) that has been validated in mice and primates was calculated using morphometric measures and physical activity as in humans [6,23]. The variables used were total distance traveled (cm), maximum speed (cm/s), mean speed, and body mass index (g/cm^2^), and they are expressed in Appendix A as the mean ± SD. To calculate the frailty index, the youngest group of 2 months old was used as a reference, and the following steps were followed for each variable: the data that differed by ±1 SD from the reference group were assigned a frailty index of 0.25; values exceeding ±2 SD were assigned a frailty index of 0.5; those differing by ±3 SD were assigned a value of 0.75; and those exceeding ± 3 SD received the maximum value of 1. The frailty indices of each variable were summed and divided by the number of variables, resulting in the final frailty index.

### 4.4. Gene Expression Analyses

Total RNA was extracted from skeletal muscle with TRI Reagent™ Solution (Thermo Scientific, Madrid, Spain) and electrophoresed in 1.5% agarose to check for RNA integrity. Total RNA was quantified in a NanoDrop by 260/280 nm absorbance, and reverse transcription was performed using a qScript cDNA Synthesis Kit (Quantabio, Beverly, MA, USA). Amplification was performed using quantitative real-time PCR in a Stratagene Mx3005P QPCR System (Agilent Technologies, Barcelona, Spain) according to the standard curve method with the PerfeCTa SYBR Green FastMix Low ROX (Quantabio, Beverly, MA, USA). gapdh housekeeping was used as an endogenous reference gene. Table 1 shows the primers used in the analysis.

### 4.5. Skeletal Muscle Histology and Morphometric Analysis

Zebrafish samples were fixed for 48 h by embedding them in 10% buffered formalin. After proper fixation, the fish samples were dehydrated by passing in ascending graded ethanol concentrations. The fish were then cleared in xylene and embedded longitudinally in paraffin. The paraffin blocks were trimmed until they reached the median plane of the fish, where longitudinal 4 µm thick sections of skeletal muscles were cut and mounted on glass slides. Following deparaffinization with xylene, the tissue sections were stained with hematoxylin and eosin (H&E) and Van Gieson (VG) stains to characterize skeletal muscle architecture and differentiate connective tissue and muscle fibers. All staining procedures were performed as described by Bancroft and Gamble. The sections were then mounted, covered, and examined, and digital images were obtained using a Carl Zeiss Primo Star Optic Microscope and a Magnifier AxioCam ICc3 digital camera (BioScience, Jena, Germany).

Morphometrical analyses of zebrafish skeletal muscle, including the number and width of myocytes, in addition to the percentage of collagen content, were performed on histological images using ImageJ 1.53 software. The analysis of the collagenous tissues was conducted on the images of Van Gieson-stained sections at a ×40 magnification, and the percentage of collagen content was calculated in the microscope field using the following formula:$$\left(\frac{Area\;occupied\;by\;collagen}{Total\;field\;area}\right)\times 100$$

### 4.6. Transmission Electron Microscopy Analysis

Small pieces of skeletal muscle were dissected from the lateral part of the trunk region and rapidly fixed in 2.5% glutaraldehyde in 0.1 M cacodylate buffer (pH 7.4), followed by post-fixation in 0.1 M cacodylate buffer with 1% osmium tetraoxide and 1% potassium ferrocyanide for 1 h. The specimens were immersed in 0.15% tannic acid for 50 s, incubated in 1% uranyl acetate for 1.5 h, dehydrated in ethanol, and then embedded in resin. Ultrathin sections were cut using a Reichert-Jung Ultracut E ultramicrotome, stained with uranyl acetate and lead citrate, and finally examined using a Carl Zeiss Leo 906E electron microscope.

Morphometrical analyses including length of sarcomeres, number, and cross-sectional area (CSA) of intermyofibrillar mitochondria (IFM), as well as number and CSA of subsarcolemmal mitochondria (SSM), were performed using TEM images and ImageJ 1.53 processing software. These parameters were measured on an area with a 5.15 µm width and 5.15 µm height. The damaged mitochondria, characterized by features such as swelling, vacuolation, depleted and/or disorganized cristae, were quantified. The percentage of mitochondrial damage was calculated in the measured area using the formula: $$\left(\frac{Number\;of\;damaged\;mitochondria}{Total\;mitochondrial\;number}\right)\times 100$$

Morphometrical analyses were conducted on five randomly selected sections of the skeletal muscle per fish. 

### 4.7. Confocal Microscopy Analysis

Skeletal muscle was extracted from Mito-GFP zebrafish and fixed with 4% paraformaldehyde (PFA) for 5 h at 4 °C and then washed in phosphate-buffered saline (PBS) for 10 min. Then, the skeletal muscle was incubated in 30% sucrose, which acts as an antifreeze, and washed again in PBS. The samples were mounted in optimal cutting temperature (OCT) medium, previously frozen in cold isopentane, which was cooled with liquid nitrogen and kept at −20 °C.

The skeletal muscle was cut longitudinally in 10µ sections in the cryostat, and the sections were recovered on superfrost microscope slides, which were kept at 4 °C. Then, the samples were rehydrated in PBS for 10 min at RT and prepared with UltraCruz Aqueous Mounting Medium with DAPI (sc-24941, Santa Cruz Biotechnology, Dallas, TX, USA). The images shown correspond to 1-micron-thick confocal sections obtained using a confocal laser scanning microscope at ×120 and ×200 magnifications (Nikon A1 confocal microscope, Centro de Instrumentación Científica, University of Granada, Granada, Spain). The fluorescence was quantified as the mean intensity of GFP / number of DAPI using the ImageJ program.

### 4.8. Statistical Analysis

Statistical analyses were performed using GraphPad Prism 6 software (GraphPad, Software, Inc., La Jolla, CA, USA). Data are expressed as the mean ± S.E.M, and one-way ANOVA with Tukey’s post hoc test was used to compare the differences between the experimental groups. A *p* value of 0.05 was considered statistically significant.

## Figures and Tables

**Figure 1 ijms-25-06166-f001:**
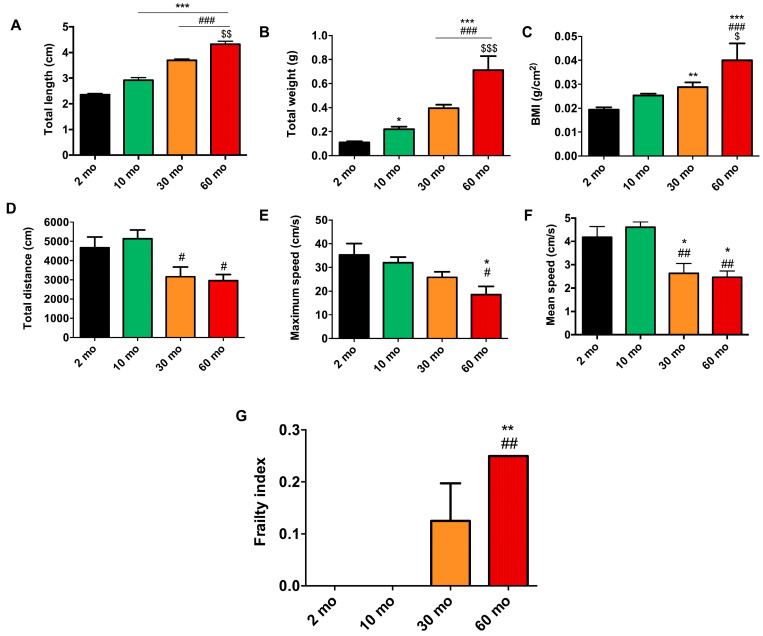
Increase in frailty index with age. (**A**) Total length (cm) and (**B**) total weight (g) increased in an age-dependent manner. (**C**) BMI (g/cm^2^) increased significantly in 30- and 60-month groups. (**D**) Total distance (cm) decreased significantly in 30- and 60-month groups with respect to the younger groups. (**E**) Maximum speed (cm/s) underwent a tendency to decrease, being significant in the oldest group with respect to the 2 and 10 month-groups. (**F**) Mean speed (cm/s) showed a prominent decrease in the 30-month group and remained in the 60-month group relative to the young groups. (**G**) The groups of 2 and 10 months had a frailty index of 0. However, at 30 months, we observed an increase, which became significant at 60 months. Data are presented as mean ± SEM (*n* = 8–10 animals/group). * *p* < 0.05 vs. 2 mo; ** *p* < 0.01 vs. 2 mo; *** *p* < 0.001 vs. 2 mo; # *p* < 0.05 vs. 10 mo; ## *p* < 0.01 vs. 10 mo; ### *p* < 0.001 vs. 10 mo; $ *p* < 0.05 vs. 30 mo; $$ *p* < 0.01 vs. 30 mo; $$$ *p* < 0.001 vs. 30 mo. One-way ANOVA with a Tukey’s post hoc test.

**Figure 2 ijms-25-06166-f002:**
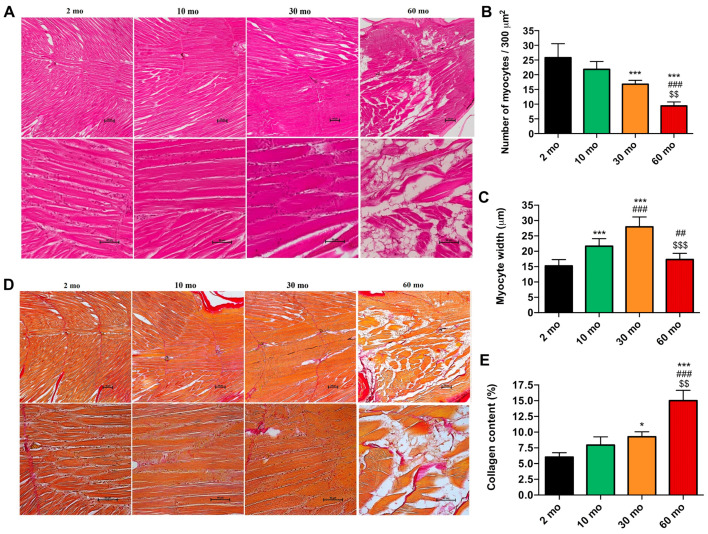
Histological and morphometric changes of zebrafish skeletal muscle. (**A**) Light microscopy images of longitudinal sections stained with hematoxylin and eosin (H&E) displayed a normal structure in the 2- and 10-month groups, whereas in the 30-month group, we observed hypertrophy and, at 60 months, muscle disorganization and atrophy. (**B**) The number of myocytes decreased with age. (**C**) The width of the myocytes increased at 10 and 30 months. (**D**) Light microscopy images of longitudinal sections stained with Van Gieson (VG) highlighted collagen infiltrations in red. (**E**) Collagen content exhibited a significant increase in 60-month-old fish. (**A**,**D**): scale bar = 100 μm (above) and 50 μm (below). Data are presented as mean ± SEM (*n* = 4 animals/group). * *p* < 0.05 vs. 2 mo; *** *p* < 0.001 vs. 2 mo; ## *p* < 0.01 vs. 10 mo; ### *p* < 0.001 vs. 10 mo; $$ *p* < 0.01 vs. 30 mo; $$$ *p* < 0.001 vs. 30 mo. One-way ANOVA with a Tukey’s post hoc test.

**Figure 3 ijms-25-06166-f003:**
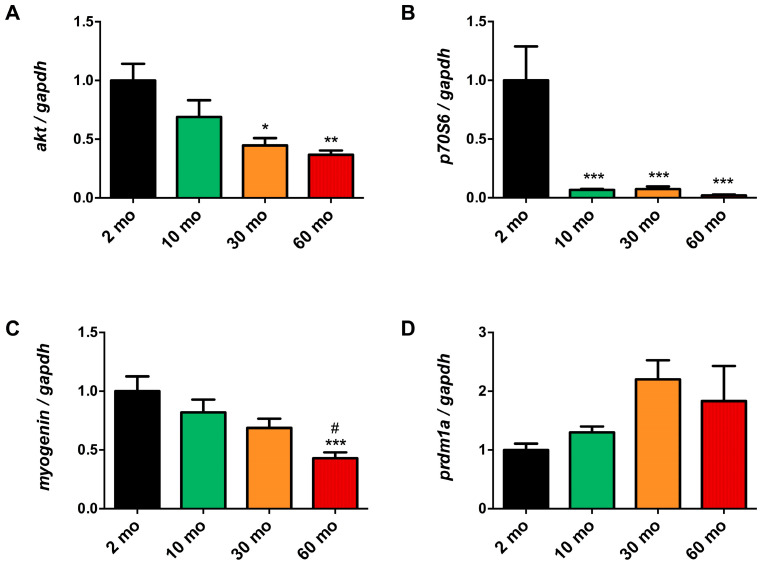
Disruption of muscle growth and differentiation pathways during aging. (**A**) akt expression showed an age-dependent decrease. (**B**) p70s6k was significantly reduced as early as 10 months of age. (**C**) Myogenin expression demonstrated a progressive reduction, with statistical significance observed at 60 months. (**D**) prdm1a increased at 30 and 60 months but not statistically significantly. Data are presented as mean ± SEM (*n* = 6–8 animals/group). * *p* < 0.05 vs. 2 mo; ** *p* < 0.01 vs. 2 mo; *** *p* < 0.001 vs. 2 mo; # *p* < 0.05 vs. 10 mo. One-way ANOVA with a Tukey’s post hoc test.

**Figure 4 ijms-25-06166-f004:**
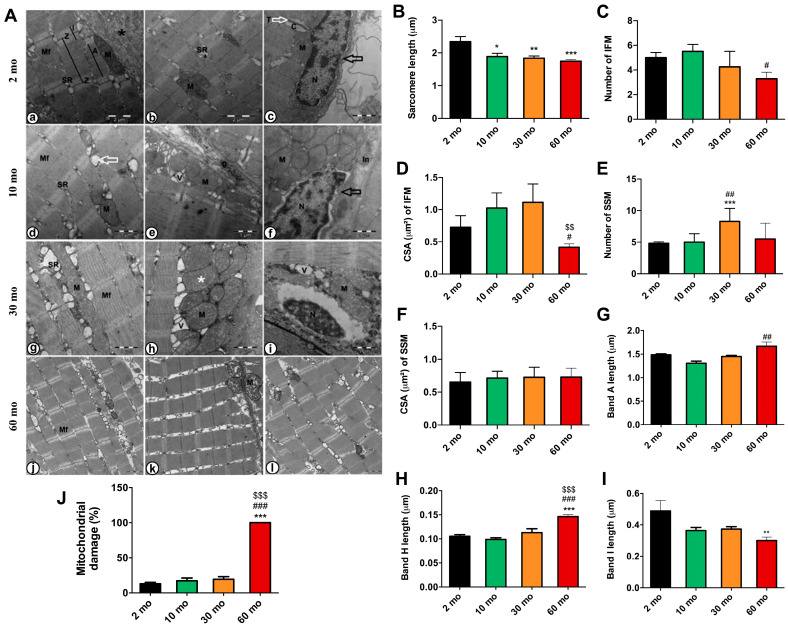
Changes in muscle ultrastructure and mitochondria. (**A**) Electron micrograph of a longitudinal section of the skeletal muscle. (**a**) 2-month-old fish showed myofibrils (Mf), isotropic bands (I), anisotropic bands and (A) sarcomeres between each two successive Z-Line (Z). (**b**) sarcoplasmic reticulum (SR), mitochondria (M) and an area of ill-developed myofibrils was detected (black asterisk). (**c**) the triad region, formed by transverse tubules (T and white arrow) and terminal cisterns (C), was evident. The nucleus (N) was peripherally located under the sarcolemma (black arrow). (**d**) At an age of 10 months old, we observed normal myofibrils (Mf), sarcoplasmic reticulum (SR), and mitochondria (M), with the presence of vacuolated mitochondria (white arrow). (**e**) It was showed vacuoles (V), glycogen droplets (g), and (**f**) peripherally positioned nucleus (N) under the sarcolemma (black arrow), and interstitial tissues (In). (**g**) 30-month-old fishes presented normal myofibrils (Mf), swollen sarcoplasmic reticulum (SR), and hypertrophied mitochondria (M), (**h**) with the presence of indistinct mitochondrial cristae (white asterisk), vacuoles (V), and (**i**) shrinkage of nucleus (N). (**j**–**l**) The 60-month-old group showed disorganized myofibrils (Mf) and mitochondria (M) with the presence of damaged ones and others with damaged and/or disorganized cristae as well as vacuoles (V). Scale bar = 1 μm. (**B**) Sarcomere length decreased with age. (**C**) The number of IFMs and (**D**) their CSA decreased significantly at 60 months of age. (**E**) The number of SSMs increased in the 30-month group. (**F**) No difference was found in the CSA of SSM. (**G**) Band A length increased in 60-month-old fish, (**H**) as did the H band. (**I**) The length of the I band decreased in the 60-month group. (**J**) Mitochondrial damage increased sharply in the older group. Data are presented as mean ± SEM (*n* = 4 animals/group). * *p* < 0.05 vs. 2 mo; ** *p* < 0.01 vs. 2 mo; *** *p* < 0.001 vs. 2 mo; # *p* < 0.05 vs. 10 mo; ## *p* < 0.01 vs. 10 mo; ### *p* < 0.001 vs. 10 mo; $$ *p* < 0.01 vs. 30 mo; $$$ *p* < 0.001 vs. 30 mo. One-way ANOVA with a Tukey’s post hoc test.

**Figure 5 ijms-25-06166-f005:**
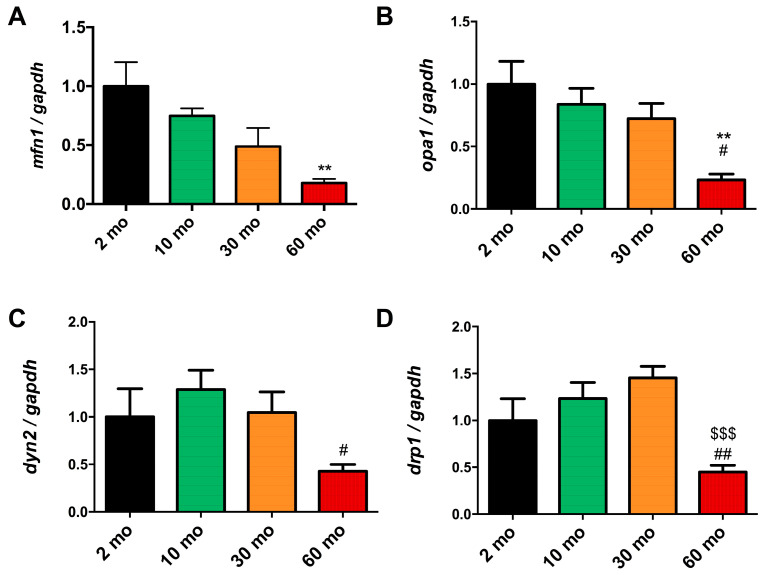
Reduction in mitochondrial dynamics in old fish. (**A**) The expression of mfn1 and (**B**) opa1 exhibited a decreasing trend with age, becoming significant at 60 months. (**C**) dyn2 expression increased at 10 months but also significantly decreased at 60 months. (**D**) The expression of drp1 increased at 10 and 30 months but showed a substantial decrease at 60 months. Data are presented as mean ± SEM (*n* = 6–8 animals/group). ** *p* < 0.01 vs. 2 mo; # *p* < 0.05 vs. 10 mo; ## *p* < 0.01 vs. 10 mo; $$$ *p* < 0.001 vs. 30 mo. One-way ANOVA with a Tukey’s post hoc test.

**Figure 6 ijms-25-06166-f006:**
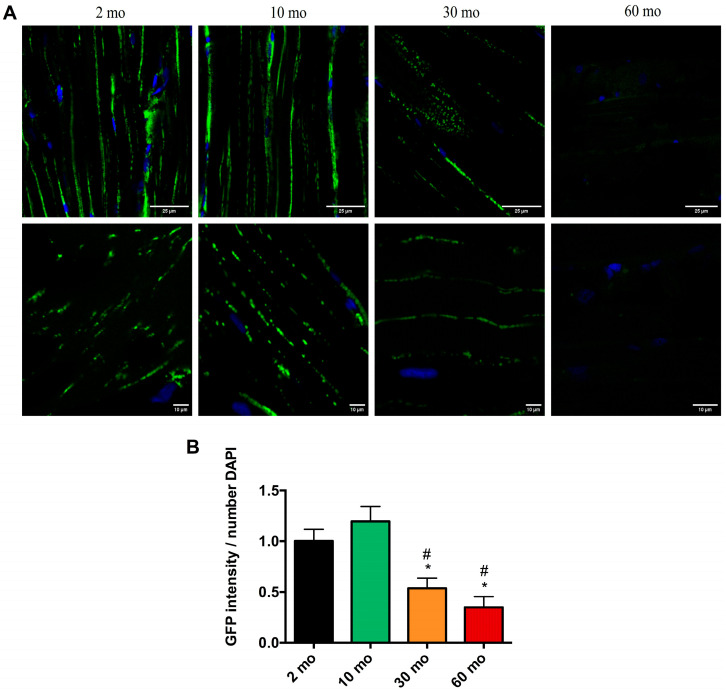
Loss of mitochondria with age. (**A**) Confocal microscopy images showing the nuclei labeled with DAPI and the mitochondria with GFP in the 2-, 10-, 30-, and 60-month groups. (**B**) The fish at 2 and 10 months exhibited a high GFP intensity, which decreased significantly at 30 and 60 months. Data are presented as mean ± SEM (*n* = 4 animals/group). * *p* < 0.05 vs. 2 mo; # *p* < 0.05 vs. 10 mo. Unpaired *t* test.

**Table 1 ijms-25-06166-t001:** Forward and reverse sequencies of the primers used for PCR.

Gene	Primer Forward	Primer Reverse
*mfn1*	AACGAAGTGTGCTCTGCTCA	GGATTCAGAGTTCGCCACCA
*opa1*	AGACTGGAAGCAGAGGTGGA	GGAAGTGACGTCGAAAGAGC
*drp1*	AACATCCAGGACAGCGTACC	TCACCACAAGTGCGTCTCTC
*dyn2*	CGCAGATAGCAGTTGTCGGA	TCTGCTTCAATCTCCTGCCG
*akt*	TGCTGAAGAGTGACGGTACG	CTTTCTTCAGGCGTCTCCAC
*p70s6k*	CAGACTCCCGTTGACAGTCC	ATTGGACTGAGAGGCGTTCG
*myogenin*	CTCCACATACTGGGGTGTCG	GTCGTTCAGCAGATCCTCGT
*prdm1a*	TTGAACGCTTTGACATCAGC	GCTGCGATGAACTTTGATGA
*gapdh*	TCACACCAAGTGTCAGGACG	CGCCTTCTGCCTTAACCTCA

## Data Availability

The datasets generated during and/or analyzed during the current study are available from the corresponding author (dacuna@ugr.es) on reasonable request. Materials described in the manuscript will be freely available for any researcher to use for noncommercial purposes.

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
