# Peer review of "Zebrafish as a Human Muscle Model for Studying Age-Dependent Sarcopenia and Frailty"

_ijms, 2024, doi:10.3390/ijms25116166_

Round 1
Reviewer 1 Report
Comments and Suggestions for Authors
Review Aranda-Martinez et al. assessed changes in muscle morphology of zebrafish at different ages and demonstrated that zebrafish is a good model to study sarcopenia. It is an interesting and well executed study and could be a valuable resource for researchers that want to start using zebrafish for their studies. However, adaptations to the manuscript are needed to improve readability and enable repetition of the experiments.
Compact, but clear introduction, personally I would only prefer a bit more information on zebrafish with respect to aging (e.g. when is zebrafish adult, life expectancy, corresponding which human age group?). Alternatively, mention this in discussion.
Results start “abrupt”, an introduction sentence of what is assessed in this first paragraph would be appreciated.
Fig1 c->b->a order would be more appropriate, as BMI is calculated based on length and weight. Adapt paragraph accordingly.
Add fold change +/- st.dev. in text when describing significant changes in e.g. speed, distance, etc. This applies to whole results section.
BMI data is compared to 2-month-old fish, but their adult at 3 months
#93/353 It is not clear how the frailty index is calculated. Results mentions that method has been adapted for zebrafish, but method/formula is not clearly explained in methods. Given the aim of the study to use zebrafish as model organism, I would expect more information on this. Value is significantly higher at 60m, but it is unclear what values mean? Also, is error bar missing or very small?
#119 specify more, number of myocytes per …. um2.
#125 VG staining? More detailed explanation in M&M of how % collagen was calculated is needed
#61+63 + gene expression section of results: full name gene (abbreviation) at first use.
#194 How is mitochondrial damage quantified?
#197 figure 4 tables are too small.
#218 number of mitochondria is reduced with aging and you subsequently analysed fusion / fission genes. Was the size of the mitochondria changed, suggesting increased fission/fusion? Analysis of mitochondrial biogenesis genes (nrf1, tfam, pgc1a) would be of added value.
#239 significant reduction in mitochondria (figure 6), but no quantification is shown. Nor is method for quantification explained in methods.
#257 “… to detect muscle” word missing in sentences?
#258-264 rephrase, first mention own results.
#271 P70S6 is downstream target of Akt, yet the gene expression profiles don’t match, as p70S6 is drastically lower already from 10m, while Akt reduction is later and less severe. Please elaborate on this.
Author Response
Compact, but clear introduction, personally I would only prefer a bit more information on zebrafish with respect to aging (e.g. when is zebrafish adult, life expectancy, corresponding which human age group?). Alternatively, mention this in discussion.
Thanks for this suggestion; we added further information in lines 69-79.
Results start “abrupt”, an introduction sentence of what is assessed in this first paragraph would be appreciated.
Corrected in each section of the results.
Fig1 c->b->a order would be more appropriate, as BMI is calculated based on length and weight. Adapt paragraph accordingly.
The figure has been changed and the paragraph adapted accordingly.
Add fold change +/- st.dev. in text when describing significant changes in e.g. speed, distance, etc. This applies to whole results section.
Values ± standard deviation have been added throughout the results section.
BMI data is compared to 2-month-old fish, but their adult at 3 months.
In the literature, it is described that the zebrafish reaches adulthood at 3 months of age because it is already sexually mature. The juvenile state (where the group of months of age would be) is defined by Parichy et al., 2009, as " the state at which most adult characteristics have been acquired in the absence of sexual maturity. Here defined by the attainment of a complete pattern of scales (squamation) and complete loss of the larval fin fold”. Therefore, fish at 2 months of age have most of the characteristics of adult fish except they lack sexual maturation.
#93/353 It is not clear how the frailty index is calculated. Results mentions that method has been adapted for zebrafish, but method/formula is not clearly explained in methods. Given the aim of the study to use zebrafish as model organism, I would expect more information on this. Value is significantly higher at 60m, but it is unclear what values mean? Also, is error bar missing or very small?
Frailty index is now explained in detail in the methods section, lines 436-449. The frailty index serves to measure the degree of vulnerability (lines 314-323 of Discussion). As can be observed in supplementary Table S1, the error of the frailty index for the 60-month group is 0.
#119 specify more, number of myocytes per …. um2.
Number of myocytes per μm2 specified (lines 143-146).
#125 VG staining? More detailed explanation in M&M of how % collagen was calculated is needed.
Explained in lines 476-479.
#61+63 + gene expression section of results: full name gene (abbreviation) at first use.
Corrected.
#194 How is mitochondrial damage quantified?
Explained in lines from 492-498.
#197 figure 4 tables are too small.
Figure 4 has been redrawed.
#218 number of mitochondria is reduced with aging and you subsequently analysed fusion / fission genes. Was the size of the mitochondria changed, suggesting increased fission/fusion? Analysis of mitochondrial biogenesis genes (nrf1, tfam, pgc1a) would be of added value.
Mitochondrial fission and fusion decrease in the 60-month-old group. As seen by TEM, there is mitochondrial damage and a reduction in the number of mitochondria in the 60-month-old fish. Therefore, one would expect mitochondrial processes such as mitochondrial fusion and fission to also be affected. It would be very interesting to study mitochondrial biogenesis as well, which should also be affected, but we cannot perform this analysis at this time because of lack of samples.
#239 significant reduction in mitochondria (figure 6), but no quantification is shown. Nor is method for quantification explained in methods.
Figure 6 has been quantified and the method explained (lines 293-302).
#257 “… to detect muscle” word missing in sentences?
Corrected.
#258-264 rephrase, first mention own results.
I don't know if I understand your comment correctly. In the legend of Figure 4, I first explain our results from section A and, then we comment the muscle structure in each experimental group.
#271 P70S6 is downstream target of Akt, yet the gene expression profiles don’t match, as p70S6 is drastically lower already from 10m, while Akt reduction is later and less severe. Please elaborate on this.
It is primarily known that p70S6K acts downstream of Akt/mTOR, but several authors have demonstrated that p70S6K is also regulated by other pathways independent of Akt/mTOR, such as the Ras/Raf/MEK/MAPK pathway (McCubrey et al., 2011; Lehman & Gomez-Cambronero, 2002; Song et al., 2005) or inhibited by the autophagy regulator ULK1 (Lee et al., 2007). Some of these Akt-independent pathways might be early affected during aging than the Akt pathway, which could explain why p70S6K is affected earlier. To better understand this, it would be necessary to study other pathways involved in skeletal muscle growth.
Reviewer 2 Report
Comments and Suggestions for Authors
Summary
This study proposes zebrafish as a model for sarcopenia. Physiological parameters including locomotor activity were significantly associated with aging, and subcellular morphology of myofibers and mitochondria also changed as observed in mammals. Expression levels of sarcopenia-related genes were also shifted with aging. These data demonstrate that zebrafish can be used as a model for sarcopenia. It’s a very exciting article to present a novel experimental system to study sarcopenia.
I strongly suggest the authors to explain the specific advantages, benefits, and usefulness of zebrafish compared to other model animals such as mice and C. elegans, which is important information for the researchers who are not familiar with zebrafish. Similarly, the methods should be described in more detail, as noted below. These revisions will make the manuscript perfect for publication in the International Journal of Molecular Sciences.
Comments
1. The number of zebrafish (or samples) for each experiment must be described.
2. Line 66: The “numerous advantages” of zebrafish should be briefly explained to emphasize the importance of this study.
3. Describe the detailed method for calculating the zebrafish frailty index so that other researchers can use the index. It’s very important for this study to establish zebrafish as a model of sarcopenia.
4. Figures 2A, 2D, and 4A: The lengths of the scale bars are hard to see. This should be noted in the figure legend.
5. Figure 4J: How did the authors judge whether the mitochondria were damaged or not? This must be described in the Materials and Methods.
6. Line 228: The legend of Figure 5A should be described.
7. The roles and/or functions of mfn1, opa1, dyn2, and drp1 are better to be explained briefly in the lines 218-226.
8. Figure 6 should be quantified and statistically analyzed (i.e., GFP signal intensity/number of DAPI).
9. I strongly recommend to quantify the mRNA levels of MuRF-1 and atrogin-1, which are the standard markers of sarcopenia. And also, age-related inflammatory markers such as TNF-α or something like that would strengthen the scientific impact of this study. These data will provide evidence that the intracellular molecular mechanisms during the progression of sarcopenia are also similar between zebrafish and other animals.
Minor points
10. Line 125: “VG” should be defined.
Author Response
- The number of zebrafish (or samples) for each experiment must be described.
The number of fish for each experiment has been described in the corresponding figure legend.
- Line 66: The “numerous advantages” of zebrafish should be briefly explained to emphasize the importance of this study.
Explained in lines 69 to 85.
- Describe the detailed method for calculating the zebrafish frailty index so that other researchers can use the index. It’s very important for this study to establish zebrafish as a model of sarcopenia.
Frailty index is now explained in detail in the methods section, lines 436-449. The frailty index serves to measure the degree of vulnerability (lines 314-323 of Discussion).
- Figures 2A, 2D, and 4A: The lengths of the scale bars are hard to see. This should be noted in the figure legend.
Scale bars are now specified in the corresponding figure legends.
- Figure 4J: How did the authors judge whether the mitochondria were damaged or not? This must be described in the Materials and Methods.
This information is now described in lines 492-498.
- Line 228: The legend of Figure 5A should be described.
Corrected.
- The roles and/or functions of mfn1, opa1, dyn2, and drp1 are better to be explained briefly in the lines 218-226.
Correctd.
- Figure 6 should be quantified and statistically analyzed (i.e., GFP signal intensity/number of DAPI).
Figure 6 has been quantified and statistically analyzed.
- I strongly recommend to quantify the mRNA levels of MuRF-1 and atrogin-1, which are the standard markers of sarcopenia. And also, age-related inflammatory markers such as TNF-α or something like that would strengthen the scientific impact of this study. These data will provide evidence that the intracellular molecular mechanisms during the progression of sarcopenia are also similar between zebrafish and other animals.
We agree with this point and it would be very interesting to study these markers of sarcopenia and inflammation. However, it is not possible at this time because there are no samples available to carry out the analysis.
Reviewer 3 Report
Comments and Suggestions for Authors
I have reviewed the manuscript and have the following suggestions for improvement:
- The title should explicitly mention that the study involves a human muscle model. This will make it clearer to readers and increase the specificity of your research focus.
- The results section of the abstract should include quantitative data rather than just descriptive text. Highlight the most significant findings with specific numerical values.
- The Materials and Methods section should be placed before the Results section.
- Expand the discussion to include insights into frailty and biometric results. Additionally, explain the selection criteria for the genes studied and provide an interpretation of genes that were not explored in this study but are relevant to the context.
- The following sentence should be revised for clarity and precision: “Thanks to the CRISPR/Cas9 system, multiple genetic models of sarcopenia can be generated, allowing the study of several genes as therapeutic targets [37].”
Author Response
- The title should explicitly mention that the study involves a human muscle model. This will make it clearer to readers and increase the specificity of your research focus.
Thanks for the observation; we corrected the title accordingly.
- The results section of the abstract should include quantitative data rather than just descriptive text. Highlight the most significant findings with specific numerical values.
Corrected.
- The Materials and Methods section should be placed before the Results section.
We placed the results before the Materials and Methods following the template of IJMS journal.
- Expand the discussion to include insights into frailty and biometric results. Additionally, explain the selection criteria for the genes studied and provide an interpretation of genes that were not explored in this study but are relevant to the context.
Insights into frailty and biometric results included in the discussion in lines 314-325.
Selection of genes here measured was determined by their participation in the main pathways of muscle regeneration and degeneration. We considered these genes and the most important for our study to reflect the alteration of muscle structure with age. Other genes including MuRF-1 and atrogin-1, which are also markers of sarcopenia were not analyzed at this moment.
- The following sentence should be revised for clarity and precision: “Thanks to the CRISPR/Cas9 system, multiple genetic models of sarcopenia can be generated, allowing the study of several genes as therapeutic targets [37].”
The sentence has been clarified in lines 406-408.
Round 2
Reviewer 2 Report
Comments and Suggestions for Authors
The authors revised the manuscript according to the reviewer's comments.